# Cardiorespiratory Fitness and Device-Measured Sedentary Behaviour are Associated with Sickness Absence in Office Workers

**DOI:** 10.3390/ijerph17020628

**Published:** 2020-01-18

**Authors:** Emma Drake, Maria M. Ekblom, Örjan Ekblom, Lena V. Kallings, Victoria Blom

**Affiliations:** 1The Swedish School of Sport and Health Science, GIH, 114 86 Stockholm, Sweden; maria.ekblom@gih.se (M.M.E.); orjan.ekblom@gih.se (Ö.E.); lena.kallings@gih.se (L.V.K.); victoria.blom@gih.se (V.B.); 2Department of Neuroscience, Karolinska Institutet, 171 77 Stockholm, Sweden; 3Division of Insurance Medicine, Department of Clinical Neuroscience, Karolinska Institutet, 171 77 Stockholm, Sweden

**Keywords:** cardiorespiratory fitness, physical activity, sedentary behaviour, sickness absence, office workers

## Abstract

Physical activity reduces the risk of several noncommunicable diseases, and a number of studies have found self-reported physical activity to be associated with sickness absence. The aim of this study was to examine if cardiorespiratory fitness, device-measured physical activity, and sedentary behaviour were associated with sickness absence among office workers. Participants were recruited from two Swedish companies. Data on sickness absence (frequency and duration) and covariates were collected via questionnaires. Physical activity pattern was assessed using ActiGraph and activPAL, and fitness was estimated from submaximal cycle ergometry. The sample consisted of 159 office workers (67% women, aged 43 ± 8 years). Higher cardiorespiratory fitness was significantly associated with a lower odds ratio (OR) for both sickness absence duration (OR = 0.92, 95% confidence interval (CI) 0.87–0.96) and frequency (OR = 0.93, 95% CI 0.90–0.97). Sedentary time was positively associated with higher odds of sickness absence frequency (OR = 1.03, 95% CI 0.99–1.08). No associations were found for physical activity at any intensity level and sickness absence. Higher sickness absence was found among office workers with low cardiorespiratory fitness and more daily time spent sedentary. In contrast to reports using self-reported physical activity, device-measured physical activity was not associated with sickness absence.

## 1. Introduction

Office workers are a group commonly associated with sedentary behaviour and low physical activity level during working hours [1]. The sedentary lifestyle at work may have contributed to the declining levels of cardiorespiratory fitness which have been seen in Sweden in recent years [2].

Having high cardiorespiratory fitness, being physically active, and minimising prolonged time spent in sedentary behaviour have been shown to reduce the risk of mortality and morbidity from several noncommunicable diseases [3,4,5,6] and mental ill-health [7], but the association with sickness absence is less examined. 

Sickness absence refers to an individual’s reduced capacity to work due to ill health [8]. It is thus not only related to health but also individual characteristics and factors at the workplace [9]. The individual may experience greater personal suffering in terms of lower income and social and psychological consequences, such as feeling isolated and powerless, as consequences of sickness absence [10,11]. It can also lead to financial strain on companies and society [10]. Furthermore, several short sickness absence spells have been shown to predict later long sickness absence [12] and can also predict premature mortality [13], even when controlling for health status.

Earlier studies have found low levels of physical activity to be associated with higher sickness absence due to musculoskeletal diseases [14,15,16], depressive disorders, and respiratory diseases [14]. However, physical activity at a vigorous level, but not at a moderate level, has shown an association with a lower risk of sickness absence [17,18]. It has been previously demonstrated that individuals who go from an inactive to a vigorously active lifestyle have a lower risk of subsequent sickness absence spells, and those who remained vigorously active exhibited the lowest risk [19]. However, earlier studies on physical activity in relation to sickness absence have only used subjective self-reported data on physical activity. This has been suggested as a major limitation, and the use of more valid, device-based measures of physical activity, such as accelerometers together with fitness tests, has been recommended for future studies [20]. 

In regards to cardiorespiratory fitness and sedentary behaviour in relation to sickness absence, very little is known. One study found cardiorespiratory fitness to be moderately associated with sickness absence due to noninjury musculoskeletal absence [21]. Low muscle fitness and aerobic endurance have been found to be associated with higher sickness absence among male military personnel [22]. Other studies have found no association in office workers [23]. Cardiorespiratory fitness has been shown to be related to work ability [24], which is a strong predictor of sickness absence [25].

While subjective measures of physical activity often refer to more intense-level activities, sedentary behaviour not just is the opposite, but refers to activities such as sitting or lying down and is often defined as activity that involves an energy expenditure ≤ 1.5 metabolic equivalents [26]. Henriksen et al. [27] found no association between sitting time and sickness absence among office workers. Prolonged sitting has in another study shown an association with less sickness absence among Finnish working-aged individuals [28].

The aim of this study was to investigate how cardiorespiratory fitness, device-measured physical activity, and sedentary time are related to sickness absence duration and frequency in a sample of office workers in Sweden.

## 2. Materials and Methods 

### 2.1. Study Design and Participants

Data were collected in 2016–2017 from the “Physical activity and healthy brain functions project”, which involved office workers at two Swedish private companies. The employees were invited to participate (*n* = 1971) via email and were asked to fill out a web-based questionnaire during working hours and to wear an accelerometer and an inclinometer. Additionally, they performed cognitive tests and a submaximal cycle ergometer test. After 6 months, participants were invited to answer the questionnaire once again. Exposure variables and covariates were taken from the baseline measurement, whereas the outcome variable, sickness absence, was derived from the follow-up questionnaire at the 6 month follow-up. The analytical sample consisted of 159 participants, 106 women and 53 men, after exclusion of individuals without valid data on variables used in this study (Figure 1). Ethical approval was granted by the Stockholm Regional Ethical Review Board (2016/1840-32). 

### 2.2. Measures 

Cardiorespiratory fitness (fitness) was estimated using the Ekblom-Bak submaximal cycle ergometer test [29], which uses heart rate recordings at two standardised work rates, together with the age and sex of the individual, to calculate VO_2_max. This test has been shown to provide a valid estimation of VO_2_ max for a wide variety of ages [30]. Relative values (mL per minute per kg body mass) were used in the present study.

Physical activity was measured using an ActiGraph GT3X (ActiGraph, Pensacola, FL, USA) at baseline. The participants were instructed to wear the accelerometer on the hip during daytime and on the wrist when they went to bed at night (necessary for sleep analyses which were not used in the present study). The accelerometer sampled 3-axial acceleration with a frequency of 30 Hz [31], and data were subsequently extracted as 60 s epochs [32] with a low-frequency extension filter [33]. Inclusion criteria included minimum wear time of 600 min of valid data on at least 4 days, excluding sleeping time [34]. Nonwear time was defined as a minimum of 60 consecutive minutes with no movement (0 counts per minute) with maximum 2 min of a vector magnitude between 0–200 counts per minute (cpm) [32]. As accelerometers were worn 24 h, sleep time was excluded based on individual sleep diaries. Standard times for in bed, 23:00, and out of bed, 6:00, was added for individuals without diaries or missing data in diaries. Light physical activity was set to 200–2689 cpm, moderate to 2690–6166 cpm, vigorous to 6167–9642 cpm, and very vigorous to >9642 cpm [35,36]. Vigorous and very vigorous physical activities were combined in the analyses. Average percentage of daily time (excluding sleeping time) spent in moderate-to-vigorous physical activity (MVPA), light (LIPA), moderate (MPA), and vigorous physical activity (VPA) were used in the analyses. 

Sedentary time (SED) was assessed at baseline using activPAL monitors (PAL technologies limited, Glasgow, U.K.) placed on the thigh [1]. Participants were instructed to wear the activPAL 24 h per day over one week, which was the same week the GT3X was worn. The devices were waterproofed and secured to the front of the right midthigh. The activPAL device measures the angle of the thigh and can thereby discriminate between sitting/lying down and standing. We used data for sitting/lying down to measure SED. The devices were initialised and processed using the activPAL software version 7.2.32 (PAL Technologies limited, Glasgow, UK) using references on awake time and bedtime from participants’ diaries. Standard bedtime (23:00–06:00) was added for individuals without diary data. Additional data processing was conducted using the HSC analysis program (developed by Dr. Philippa Dall and Professor Malcolm Granat, School of Health and Life Sciences, Glasgow Caledonian University). SED was expressed in percentage of day, excluding sleeping time. Assessing sedentary behaviour with activPAL and physical activity with actiGraph has been recommended in earlier research [37].

Sickness absence was derived from two questions assessed 6 months after baseline, specifically on dimensions of duration and frequency [38]. Duration was ascertained from the question “How many DAYS have you been home from work due to illness in the last 12 months?”. The response options were “Not at all”, “1–7”, “8–30”, “31–90”, and “91 days or more”. The question on frequency was “How many TIMES have you been home due to illness in the last 12 months?”. The response options were: “never”, “one time”, “2–5 times”, “6–10 times”, and “more than 11 times”. Sickness absence duration was dichotomized into 0–7 days and ≥8 days per year, and sickness absence frequency into 0–1 times and ≥2 times per year. The cutoff was set based on the distribution of sickness absence in our sample and was limited to low cutoffs for sickness absence due to the small analytical sample. 

Age (continuous), education (four categories: compulsory, upper secondary, university, higher academic education), gender (man, woman), smoking (yes, sometimes, no), and general health (very good, good, fair, poor, very poor) were based on self-reported data from the baseline questionnaire. These covariates were included based on earlier research regarding their associations with the exposures and outcomes [9,39,40,41,42,43,44,45]. 

### 2.3. Data Analysis

Descriptive statistics were presented for the overall sample but also according to high and low fitness level (high ≥ 39.9 mL/min/kg), MVPA (high ≥ 6.25% of time awake), and SED (low ≤ 60.49% of time awake) using median split. Mean values and standard deviations were provided for the continuous variables, and percentages were provided for the categorical variables. Multivariate analysis of variance (MANOVA) was used to examine statistical differences between the groups for the continuous variables and chi-square statistics for the categorical variables. 

Multiple logistic regression analyses were performed to calculate odds ratios with 95% confidence intervals to examine the association between the different independent variables assessed at baseline: fitness (mL/min/kg), percentage of day in SED, MVPA, LIPA, MPA, and VPA, and sickness absence at follow-up. Collinearity between the chosen covariates was examined and the variance inflation factor never exceeded two, indicating that multi-collinearity was not a concern in our models. 

All statistical analyses were performed in IBM SPSS Statistics Version 25 (IBM, Armonk, NY, USA).

## 3. Results

The total sample consisted of 159 individuals (66.7% women, mean age 43.0, SD = 8.3). The majority (59.1%) had a university education or higher (Table 1). The MANOVA showed that individuals with a high amount of MVPA had higher fitness (F = 4.29, *p* < 0.05) and more time in LIPA (F = 4.50, *p* < 0.05), MPA (F = 124.17, *p* < 0.001), and VPA (F = 41.29, *p* < 0.001) compared with individuals with low MVPA. Individuals with high fitness were of younger age (F = 19.34, *p* < 0.001) and were more likely to be men (*p* < 0.001), to spend more time in VPA (F = 11.48, *p* < 0.001), and to have better general health (*p* < 0.01) and less sickness absence (*p* < 0.01) compared with individuals with low fitness. Individuals with low SED had higher education (*p* < 0.001) and higher proportion of time spent in MVPA (F = 6.91, *p* < 0.01), more LIPA (F = 35.96, *p* < 0.001) and VPA (F = 4.99, *p* < 0.05), and smoked less (*p* < 0.05) compared with individuals with high SED. Missing data analyses were performed comparing the analytical sample (*n* = 159) (Figure 1) with those who were excluded due to missing values (*n* = different for each variable). The analytical sample was of higher age, had lower education, and had more days of sickness absence at baseline compared with the excluded individuals.

The results for sickness absence duration are presented in Table 2. For every mL increase in estimated VO_2_max, the odds of having ≥ 8 days of sickness absence per year decreased by 8% (OR = 0.92, 95% CI 0.87–0.96) in the unadjusted model. The relationship remained in all five presented models after sequentially controlling for age, education, gender, smoking, general health at baseline, MVPA, and SED. Neither SED nor PA at any intensity level showed a significant relationship with ≥8 days of sickness absence. 

Table 3 shows similar trends for sickness absence frequency. The odds of sickness absence ≥2 times per year decreased by 7% (OR = 0.93, 95% CI 0.90–0.97) for each unit increase in estimated VO_2_max (mL/min/kg) in the unadjusted model (Model 1), and the association remained after full-adjustment (Models 2–5). Percentage of the day in SED increased the odds of sickness absence ≥2 times per year by an odds ratio of 1.03 (95% CI 0.99–1.08) per percentage. This association was statistically significant after additional adjustment for age, education, and gender (in Model 2), smoking (Model 3), and baseline health (Model 4), but not when controlling for MVPA and fitness (Model 5). None of the intensity levels of physical activity were statistically associated with sickness absence ≥2 times per year.

## 4. Discussion

This study investigated the association between cardiorespiratory fitness, physical activity at various intensity levels, and sedentary behaviour on the outcome of sickness absence duration and frequency among office workers in Sweden. Higher cardiorespiratory fitness was significantly associated with lower odds of sickness absence assessed as both duration and frequency. Higher sedentary behaviour was associated with frequent sickness absence. No association was found between physical activity at any intensity and sickness absence. 

The present study supports two earlier studies that found fitness to be associated with sickness absence [21,22] but adds new knowledge regarding the associations for both sickness absence frequency and duration. However, the findings from Bernaards et al. [23] were not consistent with ours, potentially due to differences in assessments of fitness and methodology. Cardiorespiratory fitness has previously been found to be a stronger predictor for ill health compared with objectively assessed physical activity [46,47], which our results support. 

To the best of our knowledge, this is the first study that has found increased sedentary behaviour to be associated with a higher frequency of sickness absence. In contrast, Lallukka et al. [28] found an inverse association between self-reported sedentary behaviour and sickness absence. This discrepancy in results between studies may be due to the earlier study including other types of occupations in the population and not solely office workers. Moreover, Henriksen et al. [27] reported no association between self-reported sedentary behaviour and sickness absence measured in days (duration) among office workers, which was in line with the present study.

Earlier studies found self-reported physical activity to be associated with sickness absence [14,15,48,49,50], which the present study could not confirm with device-based measures of physical activity. Furthermore, our study did not find a statistically significant association between vigorous physical activity and sickness absence, which has been previously demonstrated [17,18,19]. These discrepancies between studies can also be explained by variations in methodology for measuring physical activity and sickness absence. Earlier studies have investigated differences using questionnaires and accelerometer data for assessing physical activity [51]. Another possible explanation is that the accelerometer method has a limited validity in assessing differences between the upper part of moderate-intensity, vigorous, and very vigorous activity, i.e., the types of activities that would affect maximal aerobic capacity. This inability is mainly due to a proprietary frequency filter reducing the signal at high intensities. Work has been published [52] with this filter removed, indicating far better accuracy. More studies are needed based on unfiltered data to assess the potential relation between vigorous activity and sickness absence and any relation to cardiorespiratory fitness. 

## 5. Strengths and Limitations 

Strengths of this study include the use of device-based measures of physical activity, cardiorespiratory fitness, and sedentary behaviour. This is especially important since it has been suggested as a limitation in earlier research in this area [20]. The inclusion of office employees from two companies may have limited the effect of the social gradient as a source of confounding since the occupational status and the physical work environment of the employees are rather similar. One limitation is the use of self-reported sickness absence, which can be prone to recall-bias, yet self-reported data on sickness absence has shown generally good agreement with sickness absence recorded from employers [38]. Furthermore, there may have been potential bias from the small population size, limiting the power of the study and increasing the risk of a type II error. Thus, conclusions drawn from these results require careful consideration. Reverse causality may have also impacted the results, because individuals who were too sick to work probably also were too sick to exercise. However, we aimed to reduce the effect of reverse causation by controlling for baseline health. Worth noting is that our sample was very physically active: 50% had greater than 6.25% of their wake time in moderate-to-vigorous physical activity, which corresponded to about an hour per day. The sample may be healthier than office workers in general because individuals with ill health at the time of measurements may have been less likely to participate. Additionally, sickness absence was reported 6 months after baseline, but the question refers to sickness absence in the last 12 months. Therefore, this study may be potentially interpreted as a cross-sectional study. Further, SED and LIPA are usually strongly collinear, and having them in the same model should normally be avoided. This study does, however, use data from activPAL to measure sedentary time and data from ActiGraph to measure LIPA, which makes the variables only moderately collinear (Pearson 0.51) and could therefore be used in the same model. Mainly standing time could not be included as sedentary time from activPAL or LIPA from ActiGraph. 

## 6. Conclusions

Higher cardiorespiratory fitness is associated with lower odds of sickness absence, both in frequency and duration. Time spent sedentary is associated with higher odds of frequent sickness absence. No associations were found between physical activity at different intensity levels and sickness absence. This suggests that office workers with low cardiorespiratory fitness and more sedentary behaviour may have a higher risk of sickness absence, and interventions aiming to reduce sickness absence should target these groups. The impact of fitness on sickness absence can potentially be substantial since our results suggest an increase of the odds of sickness absence of 7–8% per unit mL/min/kg. Future longitudinal studies are needed with larger samples, together with intervention studies, to explore more about the associations between fitness, physical activity, and sickness absence. The findings presented here might motivate employers to provide incentives for their employees to become more fit and/or active, and researchers to carefully evaluate/compare such initiatives. Such evaluations should include effects on physical activity patterns, cardiorespiratory fitness, work environment, mental health, productivity, and sickness absence. 

## Figures and Tables

**Figure 1 ijerph-17-00628-f001:**
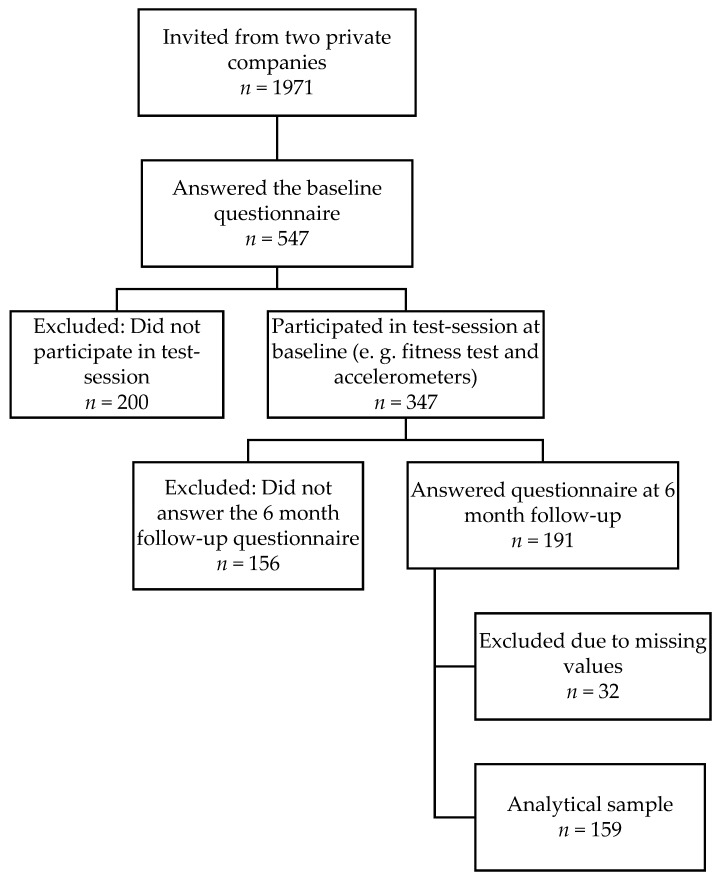
Flowchart illustrating how the analytical sample was reached from the invited individuals.

**Table 1 ijerph-17-00628-t001:** Characteristics of the sample, stratified by fitness, MVPA, and sedentary time status. Statistical differences between the mean variables were examined using MANOVA for continuous variables and chi-square tests for the categorical variables (*n* = 159).

Descriptives	ALL	High Fitness	Low Fitness	High MVPA	Low MVPA	Low SED	High SED
	*n* = 159	*n* = 80	*n* = 79	*n* = 79	*n* = 80	*n* = 79	*n* = 80
Fitness: (m ± SD) (mL/min/kg)	39.5 ± 8.6	46.5 ± 4.7 ***	32.4 ± 5.2 ***	40.9 ± 8.4 *	38.1 ± 8.7 *	40.2 ± 8.1	38.8 ± 9.1
Physical activity: (m ± SD)							
% in MVPA	6.5 ± 2.4	6.7 ± 2.5	6.2 ± 2.2	8.2 ± 2.0 ***	4.7 ± 0.9 ***	6.9 ± 2.5 **	6.0 ± 2.2 **
% in LIPA	33.0 ± 6.1	33.9 ± 5.8	32.2 ± 6.3	34.0 ± 5.9 *	32.0 ± 6.2 *	35.7 ± 5.5 ***	30.4 ± 5.5 ***
% in MPA	5.3 ± 1.9	5.3 ± 1.9	5.4 ± 1.9	6.6 ± 1.8 ***	4.1 ± 0.8 ***	5.6 ± 2.0	5.0 ± 1.6
% in VPA	1.1 ± 1.2	1.4 ± 1.4 ***	0.8 ±0.8 ***	1.7 ± 1.4 ***	0.6 ± 0.6 ***	1.3 ± 1.1 *	0.9 ± 1.2 *
Sedentary time (m ± SD) % in SED	60.8 ± 8.4	59.7 ± 7.8	61.9 ± 9.0	59.6 ± 8.0	61.9 ± 8.8	54.0 ± 5.2 ***	67.5 ± 5.0 ***
Covariates							
Age (m ± SD)	43.0 ± 8.3	40.3 ± 6.7 ***	45.8 ± 8.9 ***	42.6 ± 8.8	43.4 ± 7.9	42.3 ± 7.1	43.7 ± 9.4
Women (%)	66.7	53.8 ***	79.7 ***	69.6	63.8	70.9	62.5
Education (%)							
Compulsory education	2.5	1.3	3.8	2.5	2.5	0.0 ***	5.0 ***
Upper secondary education	38.4	32.5	44.3	34.2	42.5	25.3 ***	51.3 ***
University or equivalent	54.7	58.8	50.6	60.8	48.8	70.9 ***	38.8 ***
Higher academic education	4.4	7.5	1.3	2.5	6.3	3.8 ***	5.0 ***
Smoking yes/sometimes (%)	7.5	5.0	10.1	11.4	3.8	2.5 *	12.5 *
Lower general health (%) ^a^	17.0	8.8 **	25.3 **	11.4	22.5	16.5	17.5
Sickness absence (%)							
Duration ≥ 8 days	22.0	11.3 ***	32.9 ***	22.8	21.3	16.5	27.5
Frequency ≥ 2 times	40.3	30.0 **	50.6 **	40.5	40.0	32.9	47.5

* = *p* < 0.05; ** = *p* < 0.01; *** = *p* < 0.001. ^a^ Poor to fair health. None responded “very poor”. MVPA, moderate-to-vigorous physical activity; LIPA, light-intensity physical activity; MPA, moderate physical activity; VPA, vigorous physical activity; SED, sedentary time; SD, standard deviation; m, mean.

**Table 2 ijerph-17-00628-t002:** Odds of sickness absence ≥8 days per year according to baseline fitness, and percentage in LIPA, MPA, VPA, MVPA, and SED (*n* = 159).

Sickness Absence Duration					
	Model 1	Model 2	Model 3	Model 4	Model 5
	Odds Ratio (95% Confidence Interval)				
Fitness (mL/min/kg)	0.92 (0.87–0.96)	0.92 (0.87–0.98)	0.91 (0.86–0.97)	0.92 (0.86–0.98)	0.92 (0.86–0.98)
% in LIPA	0.98 (0.92–1.04)	0.98 (0.92–1.04)	0.98 (0.91–1.04)	0.97 (0.91–1.04)	1.00 (0.92–1.09)
% in MPA	0.97 (0.79–1.19)	0.96 (0.78–1.18)	0.93 (0.75–1.16)	0.94 (0.75–1.17)	0.96 (0.76–1.21)
% in VPA	0.80 (0.54–1.17)	0.82 (0.55–1.23)	0.83 (0.55–1.25)	0.87 (0.57–1.31)	1.04 (0.67–1.61)
% in MVPA	0.93 (0.79–1.10)	0.93 (0.79–1.11)	0.92 (0.77–1.10)	0.93 (0.78–1.12)	0.98 (0.81–1.19)
% in SED	1.03 (0.98–1.07)	1.03 (0.98–1.08)	1.02 (0.97–1.07)	1.02 (0.97–1.07)	1.00 (0.95–1.06)

LIPA, light-intensity physical activity; MPA, moderate physical activity; VPA, vigorous physical activity; MVPA, moderate-to-vigorous physical activity; SED, sedentary time. Model 1: unadjusted. Model 2: adjusted for age, education, and gender. Model 3: Model 2 + smoking. Model 4: Model 3 + general health at baseline. Model 5: Model 4 + SED, fitness and/or MVPA.

**Table 3 ijerph-17-00628-t003:** Odds of sickness absence ≥2 times per year according to baseline fitness, and percentage in LIPA, MPA, VPA, MVPA, and SED (*n* = 159).

Sickness Absence Frequency					
	Model 1	Model 2	Model 3	Model 4	Model 5
	Odds Ratio (95% Confidence Interval)				
Fitness (mL/min/kg)	0.93 (0.90–0.97)	0.93 (0.89–0.98)	0.94 (0.89–0.98)	0.94 (0.89–0.99)	0.94 (0.89–0.99)
% in LIPA	0.97 (0.92–1.02)	0.96 (0.91–1.02)	0.96 (0.91–1.01)	0.96 (0.91–1.01)	0.99 (0.92–1.06)
% in MPA	0.96 (0.81–1.15)	0.95 (0.80–1.14)	0.94 (0.78–1.12)	0.94 (0.79–1.13)	0.98 (0.81–1.18)
% in VPA	0.89 (0.67–1.18)	0.89 (0.66–1.19)	0.90 (0.67–1.21)	0.92 (0.68–1.25)	1.07 (0.78–1.46)
% in MVPA	0.95 (0.83–1.09)	0.94 (0.82–1.09)	0.94 (0.81–1.08)	0.94 (0.82–1.09)	1.00 (0.86–1.16)
% in SED	1.03 (0.99–1.08)	1.04 (1.00–1.09)	1.04 (1.00–1.08)	1.04 (1.00–1.08)	1.03 (0.99–1.08)

LIPA, light-intensity physical activity; MPA, moderate physical activity; VPA, vigorous physical activity; MVPA, moderate-to-vigorous physical activity; SED, sedentary time. Model 1: unadjusted. Model 2: adjusted for age, education, and gender. Model 3: Model 2 + smoking. Model 4: Model 3 + general health at baseline. Model 5: Model 4 + SED, fitness and/or MVPA.

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
