# Peer review of "Cardiorespiratory Fitness and Device-Measured Sedentary Behaviour are Associated with Sickness Absence in Office Workers"

_ijerph, 2020, doi:10.3390/ijerph17020628_

Round 1

Reviewer 1 Report

Thank you for the opportunity to review this article about cardiorespiratory fitness, physical activity, and office absence due to illness. This is an interesting topic and the authors have attempted to add to the knowledge base by using objective measures of physical activity through accelerometry.

Introduction: Concise and includes relevant information and clear explanation for the current study. Grammar needs to be checked throughout. ‘Sickness absence’ is not a term usually used in North America, illness or absence due to illness, would be a better term.

The justification for objectively measured physical activity is clear, the reason to assess cardiorespiratory fitness and the link to office absence is not as apparent. It appears the link is to overall health and mortality.

Materials and Methods

Data was collected in a large study 2016-17, however it is not clear why only a small percentage of the population were included in this analysis. The flow chart provides a timeline for the questionnaires but not the physical activity or fitness data collection.

Cut points are not listed, though the reference is it will be easier for the reader to have cut points listed in the article.

Is sickness absence only reported by participant on the questionnaire? How was this validated? Were records from the company checked?  The questionnaire response for Days and Times off appear very broad and only the first two categories in Days Off is not associated with high sickness absence (8 days + which includes all but the first two category which are not at all and 1-7), it appears that minimal and moderate absence is not analyzed.

What is the definition of illness on the questionnaire? Does this include illness of child or other family member? Are physical, mental, and emotional illness included? Please clarify.

Data Analysis

How were the cut-offs for high/low fitness, MVPA, and SED decided? From the article it appears they were adopted to create equal groups. What is the justification that high MVPA is equal to or greater than 6.25% of waking time? I believe more explanation would be helpful.

Results

Very interesting results. As smoking and general health are included the impact of these on fitness and physical activity should be addressed earlier in the article. Is it important to include these variables?

Discussion and Conclusions

The association between cardiorespiratory fitness and absence due to illness is well substantiated, the fact that time in physical activity did not have an association is interesting and explained well. I believe the discussion is relevant and explains the findings and suggests a practical application.

Reviewer 2 Report

This research was well conceptualized and executed. The report or article is well written. However, there are minor corrections:

Page 3; line 99- it appears that Physical Activity (PA) is supposed to be a sub-heading or that it needs to be followed by a colon. It helps to check on it because it is not fitting in the sentence. On page 7 under discussion, line 216-2020: This paragraph is clumsily written and requires rephrasing to make sense. On page 7 under discussion, line 221, "Earlier studies found...…."have" before found can be done away with as it is redundant. Page 7, line 245, "Individuals who are too sick...….delete whom and replace with who.....  

Reviewer 3 Report

This cross-sectional study examined whether cardiorespiratory fitness, device-measured PA and SB were associated with sickness absence in office workers. This study improves on previous related research by using accelerometers to measure PA and SB rather than using questionnaires. This is an interesting study as providing evidence to suggest that cardiorespiratory fitness, PA and/or SB are associated with sickness absence provides important evidence when trying to encourage employers to facilitate helping their workers to become more fit and active. However, some issues need to be addressed.

Abstract

Line 14: Typo, should be “reduces”.

Introduction

General comment: I feel this section is a little disjointed and could flow better. You mention fairly early on about sickness absence but then do not define It until the penultimate paragraph between Lines 58 and 65. I would move this paragraph so that it comes sooner in the introduction. The paragraph before the study aim should state where there is a lack of research / understanding around sickness absence and CRF, PA and SB.

Line 33: Typo, should be “Office workers are a group…”.

Line 41: Typo, should be “found low levels of…”.

Lines 55-56: I would reword this sentence: “Prolonged sitting has even shown an inverse…”.

Line 63: Typo, should be “predicts”.

Methods

General comment: You have decided to use

Line 99: Clarify when PA and SB were measured (i.e. baseline) rather than at 6 months after baseline. I would also reference this paper as you are using different measures of PA and SB to justify why you are using two activity monitors instead of one: Pfister T, Matthews CE, Wang Q, et al. Comparison of two accelerometers for measuring physical activity and sedentary behaviour. BMJ Open Sport Exerc Med 2017;3:e000227. doi:10.1136/bmjsem-2017- 000227

Line 100: Moved to the wrist for comfort? Please clarify this.

Lines 101-105: There needs to be references to support all of these decisions. Also, it looks like you have used an adapted version of the Troiano 2007 wear-time algorithm. Is this correct? Why did you not use the Choi 2011 wear-time algorithm instead? In my experience, using Troiano 2007 can remove lots of relevant SB time as it can mistake this as non-wear time. I would have concerns this may have impacted on your analysis. Please justify this decision.

Line 106: Did all participants fill in their sleep diaries? Please clarify this.

Line 107: The Aguilar-Farías et al (2014) reference is related to older adults, not adults. This 200CPM or more threshold for light PA is likely to be different for adults compared with older adults meaning it is not a suitable reference. This publication provides a more suitable reference: Petersen et al. (2015). Inclinometer Validation and Sedentary Threshold Evaluation in University Students. Res Nurs Health, although they use 150CPM or more as the threshold for light PA. Also, I would not recommend using the low-frequency extension filter. The Sasaki et al (2011) reference uses the normal filter. This is only recommended to be used in older adults and those with low function which does not apply to your sample. This will lead to inflated LPA and MVPA levels.

Line 121: I assume you are meaning “6 months after baseline”? Was sickness absence only assessed at 6 months after baseline or also at baseline? Also, the time frame for the question seems strange (i.e. last 12 months) whenever the baseline assessment was 6 months previously. Please clarify and justify this question wording.

Lines 133-134: Did you look for collinearity between the chosen covariates? Please clarify this.

Lines 138-140: Do you have a reference for classifying low and high fitness? Or did you essentially split the same in half? Please clarify this.

Discussion

Line 216: Typo, should be “To the best of our knowledge, this is the first study which has found increased sedentary…”.

Line 218: Typo, should be “earlier findings”

Line 219: Typo, should be “In line with Henriksen et al. [18], the present study found…”.

Lines 224-226: I would mention further about potential bias in using questionnaire (i.e. subjective) data in other studies. There are plenty of studies which have assessed this so I would include one or two for the reader to refer to.

Line 232: It would be useful to include the possible implications of this study. Might it encourage employers to provide incentives and/or facilitate their employees to become more fit and/or active (as an example)?

References

These need to be tidied up as there are inconsistencies (e.g. journal name abbreviated versus not being abbreviated, article title in capitals versus no capitals, different page number styles).

Round 2

Reviewer 3 Report

My comments have now been addressed by the authors.